# Ultrafast coherence transfer in DNA-templated silver nanoclusters

Erling Thyrhaug[1,†], Sidsel Ammitzbøll Bogh[2], Miguel R. Carro-Temboury[2], Charlotte Stahl Madsen[3], Tom Vosch[2] & Donatas Zigmantas[1]

DNA-templated silver nanoclusters of a few tens of atoms or less have come into prominence over the last several years due to very strong absorption and efficient emission. Applications in microscopy and sensing have already been realized, however little is known about the excited-state structure and dynamics in these clusters. Here we report on a multidimensional spectroscopy investigation of the energy-level structure and the early-time relaxation cascade, which eventually results in the population of an emitting state. We find that the ultrafast intramolecular relaxation is strongly coupled to a specific vibrational mode, resulting in the concerted transfer of population and coherence between excited states on a sub-100 fs timescale.

[1] Chemical Physics and NanoLund, Lund University, Box 124, 22100 Lund, Sweden. [2] Department of Chemistry, University of Copenhagen, Universitetsparken 5, 2100 Copenhagen, Denmark. [3] Department of Chemistry, University of Copenhagen, Thorvaldsensvej 40, 1871 Frederiksberg C, Denmark. † Present address: Photonics Institute, TU Wien, Gußhausstraße 27-29, 1040 Wien, Austria. Correspondence and requests for materials should be addressed to E.T. (email: erling.thyrhaug@tuwien.ac.at) or to D.Z. (email: donatas.zigmantas@chemphys.lu.se).

S mall, templated noble-metal nanoclusters (NCs) consisting of $<100$ atoms have received increasing attention over the last decade due to their potential applications in imaging, therapeutics and optoelectronics[1]. They are placed in the boundary between the colloid nanoparticle and atomic domains, and while a number of applications have been realized, their optical properties are not yet fully understood. The electronic structure and photoinduced dynamics of these NCs are therefore interesting also from a fundamental point of view. Gold NCs (AuNC) in particular have been investigated in some detail, where marked confinement effect-induced changes in the optical properties have been observed[2]. The broad and strongly red-shifted absorption bands characteristic of these NCs have been interpreted by treating the AuNCs as 'super-atom' cores with associated ligand-perturbed surface states[3–5]. These studies have been greatly aided by available crystal structures of several AuNC species[5,6], allowing for meaningful electronic structure calculations.

Similarly, templating procedures for the synthesis of $<30$ atom clusters of other noble metals, such as silver (AgNC) have been developed. Templating by DNA oligomers in particular is a reliable approach, allowing the synthesis of AgNCs of tunable size and properties. Such DNA-templated AgNCs show strong, sometimes chiral absorption bands in the visible-to-near-infrared (NIR) range[7–9], rather than the surface plasmon resonance typically observed around 400 nm for larger silver nanoparticles[10,11]. While the strong, broad absorption lines of AgNCs are somewhat analogous to their AuNC counterparts, a significant distinguishing characteristic is often highly efficient emission. Thus, the library of DNA-templated AgNCs contains spectrally tunable, photostable and strongly absorbing efficient emitters with large Stokes shifts. Particularly in microscopy settings, where background emission and scattering are significant concerns, these are ideal emitter characteristics[12–17]. At the same time the potential biocompatibility and low material cost makes these clusters highly attractive, for example, to sensor applications and bio-labelling[11,18–23].

The fundamental structural and electronic properties of AgNCs are however poorly understood. In particular, there is a lack of consensus regarding the energetic structure and nature of the optically active electronic excited states, which greatly impedes the interpretation of the spectroscopic signals, including observed lineshapes, Stokes shifts and polarization properties. To elucidate some aspects of the electronic structure and dynamics of these NCs we focus on an AgNC investigated by Petty et al.[7,24,25] and Shultz et al.[26], which has relatively well-characterized steady-state optical properties. This AgNC comprises 20 silver atoms and two $C_3AC_3AC_3TC_3A$ DNA oligonucleotides, and is here referred to as $Ag_{20}NC$. Here, we determine the phenomenological electronic structure of the cluster, and further demonstrate that excited-state population relaxation in the $Ag_{20}NC$ system is ultrafast and proceed in conjunction with transfer of vibrational coherence between excited-state surfaces.

## Results

**Optical properties of $Ag_{20}NC$.** The steady-state absorption and emission spectra of $Ag_{20}NC$ in 50/50 citrate buffer/ethylene glycol at 77 K are shown in Fig. 1, overlaid with the laser spectrum used in the ultrafast spectroscopy experiments. The absorption band is near-Gaussian, broad and largely featureless even at cryogenic temperatures, with a maximum at 770 nm, in agreement with earlier work[7,16,27]. No other bands with appreciable oscillator strength are observed in the visible range. This is typical of purified AgNCs, which feature near-Gaussian absorption bands in the visible-to-NIR, DNA-related absorption

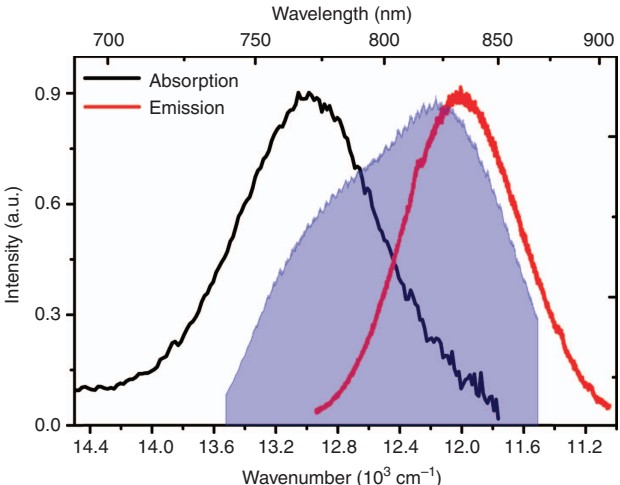

**Figure 1 | Low-temperature linear spectra of $Ag_{20}NC$.** Absorption (black) and emission (red) spectra of $Ag_{20}NC$ in 50/50% ethylene glycol and citrate buffer at 77 K. The laser spectrum used in the 2DES experiments is shown as the blue shaded area.

in the mid-ultraviolet, and minor bands in the near-ultraviolet and blue spectral regions[10,26]. Typical of colloidal samples, the quantitative absorption of the cluster is difficult to assess, as determination of the exact concentration is not straightforward[13,28]. The extinction coefficient of $Ag_{20}NC$ has however been estimated from fluorescence correlation spectroscopy to be $\sim 180,000\,M^{-1}\,cm^{-1}$ (ref. 7), which compares favourably to most laser dyes. It is worth noting that in the unpurified crude product the absorption appears as a broad semi-continuum covering most of the visible range, as has been reported in earlier studies[14,26].

On photoexcitation anywhere within the absorption band, the $Ag_{20}NC$ cluster shows strong emission in the near-infrared, centred on 835 nm. The emission is efficient in room temperature solution, with a quantum yield of 30% and an emission lifetime of 1.8 ns (ref. 7). In comparison, a number of other AgNCs have reported extinction coefficients exceeding several hundred thousand $M^{-1}\,cm^{-1}$ in combination with nanosecond decay times[10,29].

While the strong absorption and efficient NIR emission of $Ag_{20}NC$ is highly desirable in, for example, bio-imaging applications, the large Stokes shift ($\sim 1,000\,cm^{-1}$) and poor absorption/emission mirror image symmetry suggest non-trivial excited-state structure and relaxation dynamics. Little is however known about the excited-state manifold within which relaxation takes place. Theoretical treatments of the clusters are further severely hindered by the lack of an atomic structure of any DNA-templated AgNC. It is however clear that any optically induced excited-state processes must be fast, as no appreciable early-time dynamics were observed in time-correlated single photon counting (TC-SPC) experiments[7]. This implies timescales of a few tens of picoseconds or less, as expected for intramolecular processes and dynamic solvation.

To understand the early-time dynamics that ultimately shape the observed optical response, we perform a series of fully polarization-controlled two-dimensional electronic spectroscopy (2DES) experiments. Over the last decade 2DES has emerged as a highly informative tool for addressing electronic structure problems and ultrafast dynamics in complex systems. The concepts behind the technique[30] and our implementation specifically[31] are described thoroughly elsewhere. Briefly, 2DES offers several advantages over transient absorption techniques: (i)

both the absorptive and dispersive parts of the signals are measured; (ii) the optical response can be separated into rephasing and non-rephasing contributions; (iii) arbitrarily short pulses can be used without sacrificing spectral resolution, as resolution is dependent only on the homogeneous broadening of the system; and (iv) a fully spectrally resolved correlation map analogous to 2D nuclear magnetic resonance spectra can be constructed at each population time (the pump–probe delay analogue). These 2D spectra are resolved on a 'pump' ($\omega_1$) and 'detection' ($\omega_3$) frequency axis, directly revealing correlations between pumped and probed transitions. In 2D maps, the diagonal trace thus effectively contains one-colour pump–probe signals, while off-diagonal features reveal multistate correlations such as exciton delocalization and energy transfer dynamics.

**Energy-level structure.** The relaxation dynamics in $Ag_{20}NC$ are complex, thus we first establish the semi-static electronic structure from 'excited-state equilibrated' data, where fast excited-state relaxation is complete. In our experiments the cluster reaches an excited-state quasi-equilibrium after a few picoseconds and the absorptive 2D spectrum corresponding to such situation, recorded at 2 ps, is shown in Fig. 2a. The spectral contributions in this map are thus from ground-state bleach of all correlated states, and stimulated emission (SE) from the lowest equilibrated excited state only. Excited-state absorption contributions are visible at the low-energy edge of the detection window. The strongest diagonal feature corresponds directly to the broad band observed in the linear absorption data in Fig. 1, as expected. At lower energy, however, an additional diagonal feature is clearly visible. This weaker feature is found at $\sim 12,000\,cm^{-1}$ (835 nm), and corresponds to the cluster emission frequency. In the following, we refer to these absorptive transitions as the high-energy band (**H**) and the low-energy band (**L**), respectively. While low-energy states may be expected, the observation of the corresponding feature on the diagonal shows that it can be populated directly from the ground state (that is, a one-colour pump–probe signal can be observed at this energy). An intense below-diagonal cross-peak connects the two bands, showing that **L** is coupled to **H** through an energy-transfer process proceeding on an ultrafast timescale. The broadness of the bands impedes further in-depth analysis of the energy-level structure.

A common and straightforward approach to analyse broad and congested spectra is to exploit the selectivity in polarization-controlled measurements. An approach that has previously been successfully applied in 2D spectroscopy[32–34] is to construct the 'cross-peak-specific' (CP) 2D map (Supplementary Note 1 for details). In this map, diagonal spectral features (and features polarized parallel to the diagonal) are suppressed, while features

originating from 'pump–probe' interactions of pairs of states with large angles between their transition dipoles are enhanced. The $Ag_{20}NC$ CP map at 2 ps population time is shown in Fig. 2b. As expected the diagonal contribution from **L** is suppressed, while the cross-peak appears with large amplitude, implying a relatively large angle between the transitions **L** and **H**. In the **H** region, on the other hand, significant signal amplitude remains on the diagonal, and a cross-peak shifted $\sim 100\,cm^{-1}$ below the diagonal becomes apparent.

The 2D anisotropy map, shown in Fig. 2c, provides further insight. We measure and construct this map in a manner entirely analogous to pump–probe or fluorescence anisotropy measurement. Note that we only plot values in regions of significant signal amplitude, as the anisotropy is not defined in regions with no absorption (Supplementary Note 1). The **L** peak again appears with high and well-defined polarization, in accordance with the CP map. This is in good agreement with single-molecule fluorescence measurements[27,35] where the emission dipole was found to have a well-defined direction.

Conversely, regions of anisotropy values substantially lower than 0.4 are found within **H**—both at the cross-peak and on the diagonal itself. These regions of low anisotropy are present already at the very earliest times, before the **H**–**L** transfer is complete (Supplementary Fig. 1), and can thus not be the result of, for example, structural reorganization. Such deviations of the anisotropy from 0.4 (in the absence of coherences and molecular tumbling) in diagonal contributions must originate from at least two overlapping (non-parallel) transitions accessible from the same ground state. The effect of overlapping transitions on ultrafast spectra have been investigated in some detail by Smith and Jonas[36]. Their analysis is generally applicable here as well, although it is worth noting that only ground-state bleach contributions are relevant in the present case due to rapid energy transfer out of **H**. The implication is that **H** (that is, the main absorption band) does not originate from a single broadened transition, but rather from at least two transitions to distinct electronic states. We schematically summarize the emerging phenomenological electronic structure in the Jablonski diagram in Fig. 3a.

**Relaxation dynamics.** Having the phenomenological electronic structure in place, the relaxation dynamics within this structure may be characterized in detail. After photoexcitation, relaxation in $Ag_{20}NC$ proceeds by a sub-picosecond initial 'cooling' process characterized by rapid spectral shift and strong oscillatory behaviour. Following these ultrafast processes, the nanosecond ground-state recovery dominates the dynamics. This sequence of events is schematically illustrated in Fig. 3a. The ground-state recovery and other slow dynamics have been investigated

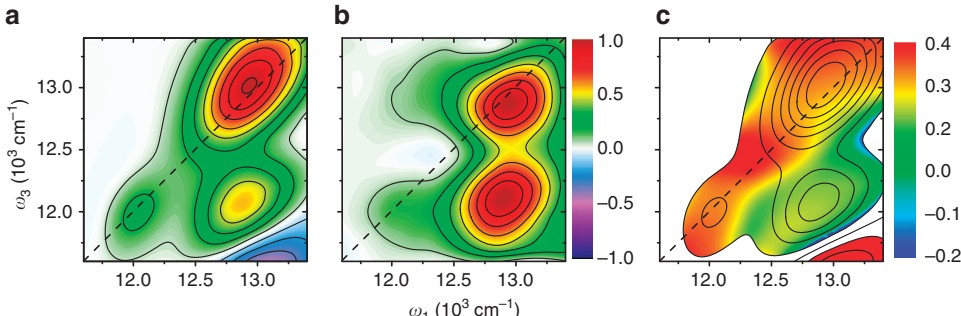

**Figure 2 | Absorptive 2D maps at 2 ps population time.** (**a**) Magic angle (MA) 2DES map, (**b**) cross-peak-specific (CP) 2D map, (**c**) 2D anisotropy map with contour lines from the MA map in **a**. The colour scale spans the range of anisotropies in the absence of coherences. The plot is masked to the area with signal amplitude of >5% of the maximum amplitude (see text for details). The spectra in **a**,**b** are normalized.

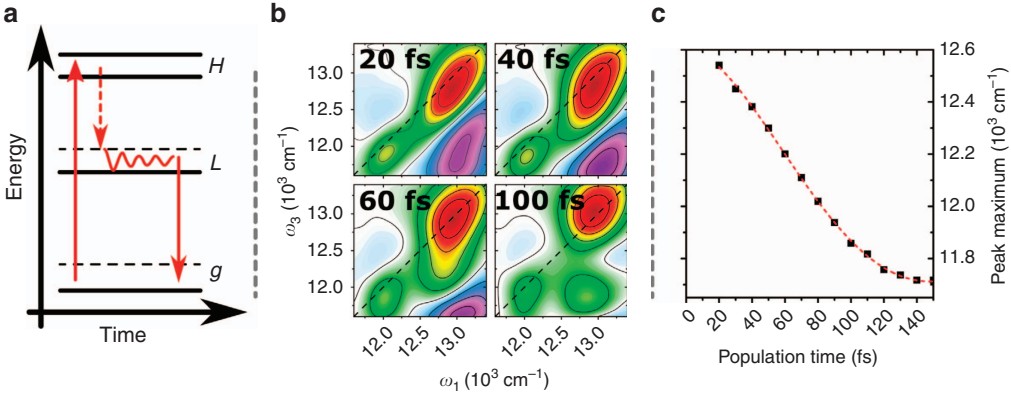

**Figure 3 | Population relaxation dynamics.** (**a**) Energy diagram of $Ag_{20}NC$ with a schematic illustration of the excited-state relaxation pathway. After excitation into the double-band-structured **H** population is rapidly transferred to **L**, followed by ground-state recovery on a nanosecond timescale; (**b**) 2DES spectra of $Ag_{20}NCs$ at short population times; (**c**) detection frequency $\omega_3$ position of the cross-peak maximum at excitation frequency $\omega_1 = 12,800\,cm^{-1}$. The fit, shown only as a guide to the eye, is a damped sine function (see text).

elsewhere in detail[7], thus here we focus on the initial relaxation processes. The 'snapshots' of early population times in Fig. 3b demonstrate several features of the early-time dynamics of $Ag_{20}NC$, where substantial time-dependence of peak positions and lineshapes is apparent (a detailed view of the sub-picosecond spectral dynamics is shown in Supplementary Movie 1 and Supplementary Fig. 2).

Several stages of relaxation can be observed after high-energy excitation: the intra-band below-diagonal cross-peak observed within **H** in Fig. 2b contains substantial SE contributions already within the excitation pulse, demonstrating that equilibration between the **H** states is ultrafast. Subsequently, the excited-state population rapidly relaxes, with the appearance of a continuous 'sliding' of the cross-peak on the energy landscape towards **L** rather than discrete state-to-state transfer. This process is complete in $\sim 140\,fs$, after which high-amplitude damped oscillations around the equilibrium position of **H**–**L** are observed. In the typical case of discrete state-to-state population transfer, the most common approach to characterize the transfer process is to determine the functional form (typically an exponential) of the time-dependent signal intensity in the initial (**H** diagonal) and the final state (at **H**–**L** cross-peak position). We show the results from this procedure in Supplementary Fig. 3. In the $Ag_{20}NC$ system, however, the cross-peak SE feature directly reveals the energetic position of the population, thus it is more convenient to quantitatively follow the transfer dynamics by tracking the position of this feature as it evolves in time, as shown in Fig. 3c. It is not obvious which functional form this relaxation takes, as it is not *a priori* clear it should be an exponential process, and a number of functions provides satisfactory fits. A damped sinusoidal fit is provided as a guide to the eye in Fig. 3c, although, for example, sigmoidal and multi-exponential models also provide satisfactory fits. In either case, the effective 'half-life' of the transfer process is $\sim 60\,fs$—substantially faster than typical vibrational cooling and internal conversion processes observed in molecules and semiconductor systems[37].

**Quantum beats**. In conjunction with the ultrafast population transfer from **H** to **L**, high-amplitude oscillations are initiated. These quantum beats (QBs) originate from the time evolution of linear superpositions of quantum mechanical states (for example, nuclear wavepackets), induced by the short broadband pulses. QB signals have complex-valued periodic time evolution of the type $S_{QB} \propto e^{\pm i\omega t}$ in population time (omitting amplitudes, dephasing and phase shifts), where $\omega$ is the QB frequency (details of the

analysis can be found in Supplementary Note 2). Owing to the complex-valued nature of the signals measured in 2DES, including oscillations, their corresponding frequencies can be both positive and negative. Fourier transforming real-valued experimental data only (for example, corresponding to transient absorption experiments) yields the beat frequency (and thus the relevant energy-level spacing), but not the frequency sign. As QBs of both signs will always contribute to experimental signals, substantial distortions due to interferences may appear in real-valued data when these contribute to the same spectral region. This may make even simple and crucial distinctions between stimulated Raman signals and excited-state vibrational (or vibronically mixed) wavepackets difficult or even impossible[38]. 2DES is a powerful tool in the analysis of the convoluted spectral signatures of coherent superpositions of states, as the entire (complex) signal emitted by the third-order polarization is recorded. This allows for both separate analysis of the rephasing and non-rephasing response, and (through a complex Fourier transform) for individual analysis of positive and negative frequency contributions. This ensures minimal interference between the multitude of signals, facilitating reliable assignment of QBs to distinct physical processes.

To extract pure QB signals from $Ag_{20}NC$ we first subtract the non-oscillatory dynamics in each point of the 2D map. Representative purely oscillatory residuals in the absorptive (real-valued) signal, showing dephasing times of $\sim 800\,fs$, are shown in Fig. 4b. Complex Fourier transforming the residuals and integrating over the whole 2D spectrum results in an integrated amplitude spectrum (IAS), also shown in Fig. 4b. When ground-state vibrations dominate, the IAS is closely related to the resonant Raman spectrum, although the correspondence is not trivial. The IAS extracted from the rephasing data reveals that the dominant QB mode has a frequency of $\sim 105\,cm^{-1}$ and almost equal amplitude in positive and negative frequencies. QBs with similar frequencies have been observed in the ultrafast spectra of AuNCs of comparable size to $Ag_{20}NC$ (refs 39–42), and in, for example, the far-infrared spectra of zeolite-templated Ag clusters[43,44]. In these studies the QBs have been assigned to acoustic 'lattice-breathing' modes, which have a direct frequency to size relation. Thus, we can, in analogy to these earlier studies, estimate the effective cluster size from the expression[45]:

$$\tilde{\nu}_R = \frac{\eta c_l}{2\pi Rc} \tag{1}$$

Here the QB frequency is written as a function of the longitudinal speed of sound ($c_l$), the particle radius ($R$), the speed

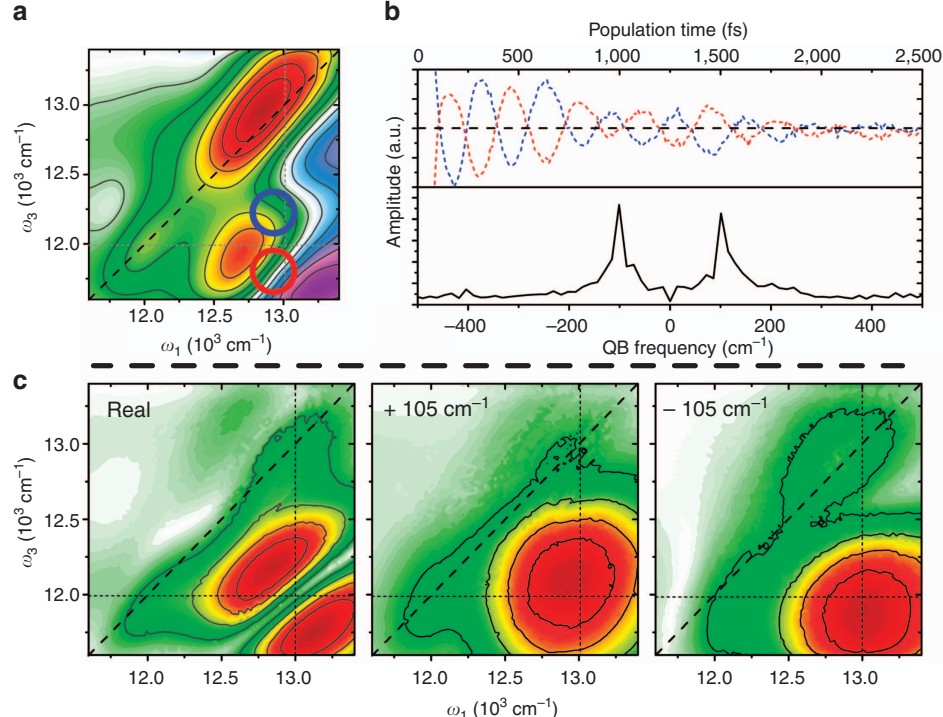

**Figure 4 | Coherent dynamics. (a)** Absorptive (real part of the) rephasing 2D spectrum at 2 ps population time. **(b)** Colour-coded residuals extracted from the circled areas in **a** (top of panel), and integrated Fourier amplitude spectrum (bottom of panel). **(c)** Rephasing QB Fourier amplitude maps. The real fourier amplitude (FT) map (left of panel) shows a characteristic nodal line. Complex fourier amplitude QB maps (middle and right of panel) show the individual frequency contributions.

of light ($c$) and an eigenvalue for breathing modes ($\eta$) that is ~2.85 for silver. From these data we can estimate an effective particle diameter of ~1.1 nm. While direct structural information is not available for $Ag_{20}NC$, this estimate is close to what one would expect given the size of, for example, $Au_{25}NC$ (ref. 40) and other characterized noble-metal clusters with similar number of atoms.

Plotting the amplitude of this mode as a 2D map of 'pump' and 'detection' frequencies yields a detailed picture of the induced wavepackets. We plot the QB amplitude map resulting from the real Fourier transform of rephasing data in the left of Fig. 4c, where the distorted shape and characteristic nodal line are apparent. Although the interference prohibits unambiguous interpretation of the map, it is clear that the dominating QB appear at the detection frequency of **L** after excitation into **H**. This is significant in that it implies unusual coherence dynamics: either the dominating coherence is a Raman active ground-state vibration that is predominantly initiated by **H** excitation but best detected around the **L** energy, or a wavepacket is generated in **H** and transferred to **L** on the same ultrafast timescale as the energy transfer process itself. Only by complete separation of the signal into rephasing and non-rephasing signals of positive and negative frequencies can we clearly assign the QBs.

The rephasing positive and negative QB amplitude maps are shown in Fig. 4c. Two intense, broad peaks appear close to the below-diagonal cross-peak, split by the beat frequency along $\omega_1$ and by twice the beat frequency along $\omega_3$. By making two observations about rephasing coherences, these maps can be used to directly assign the QBs to a physical process: (1) excited-state wavepackets will generate symmetrically displaced, similar-amplitude QBs of both frequency signs; and (2) ground-state contributions (that is, impulsive stimulated Raman scattering, ISRS) will only appear with negative sign (Supplementary Note 2 and Supplementary Figs 5–9). The observation of displaced

equal-amplitude ± frequency QBs around the **H–L** cross-peak is thus a clear signature of excited-state wavepacket motion, excluding stimulated Raman processes as major contributors to the QB signal.

In addition to the dominating coherent signal around the cross-peak, less-intense QBs are present both on the **L** diagonal (both negative and positive frequencies) and around the **H** diagonal (predominantly with negative frequency). The latter in particular demonstrates that ISRS processes also contribute to the total coherent response, in agreement with observations in AuNC studies[39,42]. In simple displaced oscillator models the amplitude of vibrational QBs is directly related to the relative displacement of the potential energy surfaces involved in the optical transition. This suggests that one can use the relative QB amplitude as a qualitative first estimate of potential energy surface displacement also in more complex systems such as $Ag_{20}NC$. For the ISRS signal in the vicinity of the **H** diagonal peak we find the amplitude to be only 2–3% of the total signal, suggesting that the **H**-band states are only slightly displaced relative to the ground state. At the **L** diagonal peak on the other hand, the relative QB amplitude is on the order of 25% of the non-coherent signal, suggesting a much larger displacement relative to the ground state. The implication is that the **H**-band to **L**-band population transfer appears to involve a transition between two relatively significantly displaced surfaces.

## Discussion

Analysis of the experimental results leads to the phenomenological energy-level structure shown in Fig. 3a for the representative silver NC $Ag_{20}NC$, and to identification of unusually fast population and coherence transfer within this structure. Interpreting the observed response in the framework of a specific electronic structure model is less straightforward, in the view of

the fact that somewhat substantially differing models have been proposed[14,35,41,46–48]. The spectral features have alternately been interpreted as arising from plasmon resonances or from transitions in a 'super-atom'. A plasmonic origin of **L** in particular is unlikely due to the gross mismatch between the state lifetime (nanoseconds) and plasmonic dephasing times ($\sim$10–100 fs or less[49]). Plasmonic contributions to **H** cannot be excluded *a priori* however, and some theoretical work has shown plasmon resonances at higher energies in low-dimensional metallic structures[50].

Notwithstanding that there are some substantial differences in the phenomenological optical response and excited-state relaxation of Au- and AgNCs, conceptually the super-atom picture used to explain the photophysics of AuNCs maps well onto the $Ag_{20}NC$ data presented here. By analogy to the transitions to a manifold of core states observed for AuNCs[3,5], **H** can originate from closely spaced intra-core transitions. Following this, the **L** band can be interpreted as transitions to localized states, possibly involving surface atoms directly bonded to the nucleobases of the DNA template. The persistent anisotropy in the signal points to a low overall electronic state symmetry, which may originate due to non-symmetrical cluster geometry or asymmetric ligation to the strongly interacting DNA nucleobases. The consequence of this low effective symmetry is a splitting of the transitions to **H**-band states into two distinct bands.

The population relaxation between the manifolds of states appears to proceed with a continuous energy-loss mechanism, resulting in smooth transfer of the SE signal from **H** to **L** with no distinguishable discrete steps. This stands in contrast to the dynamics of similar-sized AuNCs (ref. 39), where no such process has been observed, and has strong implications for the interpretation of the transfer process. While the relaxation takes a shape in qualitative agreement with expectations from systems having a (quasi-) continuous energy-level structure, processes such as, for example, carrier relaxation in semiconductors after above-bandgap excitation are typically considerably slower[37,51]. An alternative interpretation is a non-adiabatic energy transfer process, where strong coupling to a nuclear 'reaction coordinate' (for example, a vibrational or localized phonon mode) results in continuous and extremely fast population transfer along this coordinate, in analogy to the conical intersection dynamics observed in molecular systems.

The characteristic signatures of excited-state wavepacket motion being detected in **L** after excitation into **H** provides strong corroborating evidence for this interpretation, as the ultrafast transfer of not only population but also coherence implies that the energy transfer process in $Ag_{20}NC$ is strongly coupled to nuclear motion.

Reliable observations of ultrafast coherence transfer dynamics have recently been made in connection with transfer through conical intersections in molecules[52,53] and complexes[54], where the surface crossing is directly connected to such a change in nuclear coordinates. In analogy with these studies, the observed ultrafast population transfer and high-amplitude coherent signals suggest that **H** and **L** potential energy surfaces are strongly displaced and connected through a nuclear coordinate. Such a topology provides a plausible explanation for the **H**–**L** transfer rate of over an order of magnitude larger than the analogous process in AuNCs, where no excited-state coherence has been observed[39].

The importance of this 'trapping' of the population in an emissive state by an ultrafast process is uncertain. One may expect that since it allows, for example, thermalization losses within the cluster core to be efficiently out-competed, it may play a significant role in the high emission yield of silver clusters. Further work, both theoretical and experimental, is however needed to clarify whether non-adiabatic transfer dynamics is ubiquitous in the relaxation cascade of AgNCs, and to determine its importance for the observed emission properties.

## Methods

**Femtosecond spectroscopy.** Femtosecond broadband pulses were generated by feeding the 1,030 nm output of a Pharos laser (Light Conversion Ltd) to a home-built NOPA to generate an $\sim$100 nm full-width at half-maximum spectrum centred at 805 nm. The broadband pulses were attenuated to pulse energies of 3 nJ per pulse, and compressed to 14 fs using chirped mirrors and a prism compressor. Full independent control over the pulse polarizations was achieved by a combination of a quarter wave plate and linear polarizers in each beam and in the detection path. Several measurement series were performed, with each series containing between 100 and 200 population time points. Population times were scanned for up to 900 ps to observe the long-time decay, while short-time dynamics and coherent beats were investigated starting at 0 fs using regular steps of 10 fs. The coherence time was in all cases scanned from $-150$ to $+150$ fs in 1.8 fs steps, resulting in a spectral resolution of 65 cm$^{-1}$ along $\omega_3$ and 110 cm$^{-1}$ along $\omega_1$.

**Linear spectroscopy.** Linear absorption spectra were recorded on a Perkin-Elmer Lambda 1050 UV/Vis/NIR absorption spectrometer, and emission spectra were recorded on a Horiba-Yvon FluoroLog3 spectrometer equipped with a polychromator and a charge-coupled device camera.

In all measurements samples were kept in a temperature-controlled liquid nitrogen bath cryostat (Oxford Instruments ltd.) equipped with fused silica windows.

**Nanocluster synthesis.** $Ag_{20}NCs$ were synthesized according to earlier published procedures[7,26] (Supplementary Note 3). The crude product was purified by column chromatography and dissolved in a citrate buffer, where it was found to be stable for several months. All reagents and solvents were acquired from Sigma-Aldrich, and used as received without further purification.

**Data availability.** The data that support the findings of this study are available from the corresponding authors on reasonable request.

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

## Acknowledgements

E.T. and D.Z. gratefully acknowledge financing from the Knut & Alice Wallenberg foundation, the Swedish Research Council, and NanoLund. T.V., S.A.B. and M.R.C.-T. acknowledge financial support from the 'Center for Synthetic Biology' at Copenhagen University funded by the UNIK research initiative of the Danish Ministry of Science, Technology and Innovation (Grant 09-065274), bio-SYNergy, University of Copenhagen's Excellence Programme for Interdisciplinary Research, the Villum Foundation (Project number VKR023115) and the Danish Council of Independent Research (Project number DFF-1323-00352). We thank Dr Marcelo Alcocer for helpful discussion and assistance with plotting and animation, Dr David Paleček for providing access to his code for complex Fourier transform analysis, and Professor Knud J. Jensen for providing access to the HPLC apparatus necessary for sample purification.

## Author contributions

E.T. performed the experiments and analysis; E.T., T.V. and D.Z. wrote the manuscript; S.A.B., M.R.C.-T. and C.S.M. synthesized, purified and characterized the NC sample.

## Additional information

**Competing interests:** The authors declare no competing financial interests.

