## [Peer Review File · Nature Communications]

Reviewers' comments:

Reviewer #1 (Remarks to the Author):

This paper described fully polarization controlled 2DES measurements on DNA-templated Ag nanoclusters (AgNCs) at low temperature. By the observed 2D anisotropy map, the two-level model on excited states was established, which contains an initial populated H state and a relaxed long-live fluorescent state L. The detailed quantum beating analysis indicated that both population and coherence transfers from H to L states in ultrafast time scale. The work is of interest to the ultrafast community and presented in an interesting and easily comprehensible fashion. I especially like the paragraphs discussing the origin of quantum beating then confirm the transferring of coherence. The manuscript is therefore publishable, but several points should be addressed prior to publication.

My major remark is about the assignment of quantum beating. The H band is recognized as at least two overlapping transitions correspond to distinct electronic states. As they are both populated by the excitation pulses, these two electronic states might be almost degenerative. Whether or not such electronic coupling between these two states also contribute to the observed quantum beating? Meanwhile, in the view of 2DES, the oscillations are presented in both rephrasing and nonrephrasing signals, this confirms the contribution from vibrational coherence but does not indicate the absence of electronic coherence. The authors only attributed the observed beating behaviours to wavepacket motion from H to L (dominant) and an active Raman vibrational mode, which is based on the 2DES results. But do authors have any consideration or comments on the possible electronic couplings between the degenerated states of H and/or L states?

In addition, since authors stressed that the energy transfer process in Ag20NC is strongly coupled to nuclear motion, I just wonder why author didnot perform the pump-intensity dependent quantum beating to see if there is any QB period changes, and or if it is electronic coupling or just wavepacket moving? In my opinion, that there should be no oscillation be observed in L state (since L state in other name, surface trap state), the oscillation could only occurred in electronic coupled H-state, this is why the lifetime of oscillation is very short. I just doubt the cluster is over pumped, in this case, pump-intensity dependent experiments could be performed as comparison. Authors should give some discussion about this point in their revision at least.

As the 2DES shows strong quantum beating at 105 cm⁻¹, which was attributed to the wavepacket motion and Raman vibration. Did the authors perform the resonance Raman measurements to confirm or exclude the contribution from Raman vibration? What is 105cm⁻¹ vibration from? In my opinion, it is from the base vibration of cluster itself, in agreement with previous similar works reported by Goodson, and other groups. Authors may possibly make some estimation about the 105cm⁻¹ by taking Ag20 cluster as a small ball to see the intrinsic base vibration of such small ball.

Reviewer #2 (Remarks to the Author):

This manuscript reports a novel study on the electronic structure and intramolecular relaxations on the time scale of hundreds of fs in Ag nanoclusters (AgNCs) using 2D spectroscopy. The author utilized cross-peak specific 2D spectroscopy and anisotropy maps to probe the electronic structure of the sample, and they use Feynman diagrams to understand the observed coherence. The authors attribute the observed coherence to excited-state wavepacket motion and suggest that coherence transfer plays a role in the ultrafast relaxation process. The result is a novel and interesting. These experiments are also the first 2D measurements on metal nanostructures. I recommend publication after addressing the following concerns:

1) On page 7, the authors use a cross-peak specific 2D map (Fig. 2B) at 2ps to reveal the electronic structure of the AgNCs. The author demonstrated that the diagonal peak such as L is suppressed and cross peak like H-L is enhanced. Is there any explanation for the absence of

excited-state absorption below the H-L peak?

2) On page 8, the authors attributed the low anisotropy of the diagonal H peak in the 2D anisotropy map (Fig. 2C) to the overlapped transitions from the ground electronic state to multiple electronic states in H-band. The anisotropy 2D map at 2ps is shown. At 2 ps, the author described in the manuscript that the population relaxation from H to L is complete. Thus the anisotropy decay at 2 ps can also be attributed to the population relaxation. A 2D anisotropy map at early times before population relaxation would be a better support for this argument.

3) The horizontal dashed line in Fig. 2B is not at the H peak. The scale of the colorbar in Fig. 2C is truncated.

4) In the last paragraph of page 10, the author suggested the ultrafast equilibration between the H states occurs within 20 fs because of the observation of an intraband cross peak within H. It would be nice to show the 2D map in the manuscript or SI. Furthermore, if two transitions share a common ground state, the cross peak shows up before equilibrium. Is there any evidence to distinguish these two interpretations?

5) On page 10, the authors state that population relaxation from H to L is completed in ~ 140 fs.

6) On page 11, they use the cross peak maximum to track the dynamics. They should provide further justification for this approach. Why not to use the cross-peak amplitude to reveal the population relaxation? A rise of the cross-peak amplitude within 140 fs would be expected for the H-L relaxation. It would also be nice to show some 2D maps at different population times and time traces of diagonal and cross peak amplitudes to reveal the population relaxation dynamics.

7) The authors present a clear discussion of the observed coherence and attribute it to vibrational wavepacket motion in the excited electronic state. Have there been previous studies on similar samples that observe similar vibrational modes, perhaps from resonance Raman work?

8) On page 15, the authors claim that the coherence at the H-L peak transferred from H to L is much stronger than the coherence directly photoexcited at L. This is different from other systems in which the direct excitation of coherence in either ground or excited electronic states would be expected to be stronger than a coherence transfer from another states. In addition, the vibrational coherence in the ground electronic state is typically stronger than that in excited electronic state. The authors should comment on this in the manuscript.

Reviewer #1 (Remarks to the Author):

This paper described fully polarization controlled 2DES measurements on DNA-templated Ag nanoclusters (AgNCs) at low temperature. By the observed 2D anisotropy map, the two-level model on excited states was established, which contains an initial populated H state and a relaxed long-live fluorescent state L. The detailed quantum beating analysis indicated that both population and coherence transfers from H to L states in ultrafast time scale. The work is of interest to the ultrafast community and presented in an interesting and easily comprehensible fashion. I especially like the paragraphs discussing the origin of quantum beating then confirm the transferring of coherence. The manuscript is therefore publishable, but several points should be addressed prior to publication.

My major remark is about the assignment of quantum beating.

The H band is recognized as at least two overlapping transitions correspond to distinct electronic states. As they are both populated by the excitation pulses, these two electronic states might be almost degenerative. Whether or not such electronic coupling between these two states also contribute to the observed quantum beating?

Author Comment:

From the position of the *intra*-H-Band cross-peak we can see that the splitting of these bands is *approximately* 100 cm^{-1} . As the reviewer points out, transitions to both states are induced by the laser pulses, so quantum superposition states (electronic coherence) are necessarily created upon excitation. We expect these electronic coherences to dephase very rapidly however. The most straight-forward argument for this is:

- Quantum beats (QBs) from electronic coherence (or excited state wavepackets) require population in the relevant states. In this system all population has transferred from the H-band states to L within $\sim 150\text{ fs}$, and there cannot be any electronic coherence persisting within the H-band on timescales longer than this.

Direct excitation of the coherence between H-band L states could in principle contribute to the H-L cross-peak, however these can also be excluded as the QB would appear with the corresponding frequency difference – *approximately* 1000 cm^{-1} .

On the basis of these arguments we exclude the possibility that “pure” electronic coherence contributes substantially at times longer than $\sim 100\text{ fs}$.

Meanwhile, in the view of 2DES, the oscillations are presented in both rephrasing and nonrephrasing signals, this confirms the contribution from vibrational coherence but does not indicate the absence of electronic coherence. The authors only attributed the observed beating behaviours to wavepacket motion from H to L (dominant) and an active Raman vibrational mode, which is based on the 2DES results. But do authors have any consideration or comments on the possible electronic couplings between the degenerated states of H and/or L states?

Author Comments:

As noted above, electronic coherence will be present at short times as a consequence of the broadband excitation, but will dephase rapidly due to ultrafast population transfer as well as strong interaction with the environment. As the reviewer implies however; a “conical-intersection-type” dynamic does imply that *at least* at the crossing-point electronic and vibrational states are no longer separable. This is a very interesting situation, and the related research into *vibronic* coherence is currently a highly active field. It is however difficult to quantify this “state-mixing” experimentally and we expect these data to present an interesting and extensive challenge to theoreticians.

In addition, since authors stressed that the energy transfer process in Ag20NC is strongly coupled to nuclear motion, I just wonder why author did not perform the pump-intensity dependent quantum beating to see if there is any QB period changes, and or if it is electronic coupling or just wavepacket moving? In my opinion, that there should be no oscillation be observed in L state (since L state in other name, surface trap state), the oscillation could only occurred in electronic coupled H-state, this is why the lifetime of oscillation is very short. I just doubt the cluster is over pumped, in this case, pump-intensity dependent experiments could be performed as comparison. Authors should give some discussion about this point in their revision at least.

Author Comment:

While we are well into the weak-excitation regime in these experiments, we agree that the pump-intensity dependence should be controlled, and to address this we have included a comparison of residuals of time traces recorded at different pulse energies in Supporting Information Figure 4. We find no QB frequency change in experiments recorded at 3 nJ/pulse (the pulse energy used for data presented in the main paper) and 1.5 nJ/pulse.

Regarding the QBs:

when excitation is into the L state around 12000 cm^{-1} we find quantum beats typical of a displaced oscillator – i.e. we observe QBs in both bleach and stimulated emission parts of the signal, in other words we measured wavepackets on ground and excited state potentials. When excitation is into H, we find weak bleach oscillations in the H-band signal (impulsive stimulated Raman) and strong oscillations in the stimulated emission signal from the L state. In either case it is however clear that both ground-

and excited-state signals are affected by the QB, which dephases in ~ 800 fs. We have expanded our discussion of the QBs and their origin in order to clarify this in the results section (pages 10-11)

As the 2DES shows strong quantum beating at 105 cm^{-1} , which was attributed to the wavepacket motion and Raman vibration. Did the authors perform the resonance Raman measurements to confirm or exclude the contribution from Raman vibration? What is 105 cm^{-1} vibration from? In my opinion, it is from the base vibration of cluster itself, in agreement with previous similar works reported by Goodson, and other groups. Authors may possibly make some estimation about the 105 cm^{-1} by taking Ag₂₀ cluster as a small ball to see the intrinsic base vibration of such small ball.

Author Comment:

We agree with this assessment. From the work of Goodson and others, the most probable assignment of the QB is a lattice breathing mode in the cluster. The suggestion of estimating the acoustic phonon frequency for a small silver ball is good, and we have included comments (and the relevant equation) on this in the Results section (pages 10-11). From the observed frequency, we find a diameter of approximately 1.1 nm. No structure exist of the cluster, however we find this simple approximation to be largely in line with observations from gold clusters. As further confirmation we include references to vibrational spectroscopy experiments on similarly sized Ag-in-Zeolite clusters, where (lattice expansion) cluster modes are observed in approximately same frequency range.

Reviewer #2 (Remarks to the Author):

This manuscript reports a novel study on the electronic structure and intramolecular relaxations on the time scale of hundreds of fs in Ag nanoclusters (AgNCs) using 2D spectroscopy. The author utilized cross-peak specific 2D spectroscopy and anisotropy maps to probe the electronic structure of the sample, and they use Feynman diagrams to understand the observed coherence. The authors attribute the observed coherence to excited-state wavepacket motion and suggest that coherence transfer plays a role in the ultrafast relaxation process. The result is a novel and interesting. These experiments are also the first 2D measurements on metal nanostructures. I recommend publication after addressing the following concerns:

1) On page 7, the authors use a cross-peak specific 2D map (Fig. 2B) at 2ps to reveal the electronic structure of the AgNCs. The author demonstrated that the diagonal peak such as L is suppressed and cross peak like H-L is enhanced. Is there any explanation for the absence of excited-state absorption below the H-L peak?

Author Response:

We observe that the ESA transition(s) causing the signal covering the below H-band region is close to parallel to the H-band transition, and as a result are suppressed in the CP specific map. Several spectra illustrating this have been included in SI figure 3. We had in the original manuscript accidentally masked off the ESA region of the anisotropy map (Fig. 2C), which may have obscured this point. This has been fixed, and Fig 2C now also shows the anisotropy of the low-energy ESA region.

2) On page 8, the authors attributed the low anisotropy of the diagonal H peak in the 2D anisotropy map (Fig. 2C) to the overlapped transitions from the ground electronic state to multiple electronic states in H-band. The anisotropy 2D map at 2ps is shown. At 2 ps, the author described in the manuscript that the population relaxation from H to L is complete. Thus the anisotropy decay at 2 ps can also be attributed to the population relaxation. A 2D anisotropy map at early times before population relaxation would be a better support for this argument.

Author Response:

The anisotropy will typically be unaffected by population transfer (but not coherence – see below) as long as the direction of the transition moments themselves do not change during the measurement (*i.e.* as long as the states themselves are unaltered), and as long as the SE from a given state is parallel to the absorption to the same state. Significant failure in fulfilling this typically requires some permanent structural reorganization in the cluster, or rotational diffusion of the cluster itself. As we do not observe any such dynamics we can take the 2 ps spectra as representative.

A concern at short times is that coherence can contribute strongly to the anisotropy (e.g., the range is no longer -0.2 to 0.4), complicating the data-analysis. It is possible to approximately correct for this by taking the coherence dephasing time and estimated relative amplitudes into account. We have included a number of early-time polarized spectra corrected in this way in SI figure 3, and refer to these in the text. These spectra lead to the same conclusions as the 2 ps spectrum.

3) The horizontal dashed line in Fig. 2B is not at the H peak. The scale of the colorbar in Fig. 2C is truncated.

Author Response:

We have updated the figures to fix the issues pointed out by the reviewer.

4) In the last paragraph of page 10, the author suggested the ultrafast equilibration between the H states occurs within 20 fs because of the observation of an intraband cross peak within H. It would be nice to show the 2D map in the manuscript or SI. Furthermore, if two transitions share a common ground state, the cross peak shows up before equilibrium. Is there any evidence to distinguish these two interpretations?

Author Response:

We can see from the polarized spectra that the H-band states indeed share a common ground-state – it is the cause of the low anisotropy on the diagonal. We have clarified the text to reflect that in addition we see the stimulated emission component appearing at the H-band cross-peak within the laser pulse. We also now explicitly point this out in the connection with the dynamics. Additionally, for illustration, we have included several more early-time spectra in the SI, as well as early-time CP spectra (SI figures 1 and 3 respectively).

5) On page 10, the authors state that population relaxation from H to L is completed in ~ 140 fs.

6) On page 11, they use the cross peak maximum to track the dynamics. They should provide further justification for this approach. Why not to use the cross-peak amplitude to reveal the population relaxation? A rise of the cross-peak amplitude within 140 fs would be expected for the H-L relaxation. It would also be nice to show some 2D maps at different population times and time traces of diagonal and cross peak amplitudes to reveal the population relaxation dynamics.

Author Response:

We have expanded the text (pages 8-9) to clarify the choice of observable for monitoring population transfer. In addition to this we have included diagonal- and cross-peak kinetics relevant to H-to-L transfer in SI figure 2 and refer to this in the text. The two approaches do not give significantly different transfer times. We have included a number of additional early time spectra (see our answer to point 4 above), and we also feel that the Supporting Movie (referred to in the text - an animation consisting of 2D spectra from 0 to 1000 fs in 10 fs steps) is a good qualitative illustration of the population transfer dynamics.

7) The authors present a clear discussion of the observed coherence and attribute it to vibrational wavepacket motion in the excited electronic state. Have there been previous studies on similar samples that observe similar vibrational modes, perhaps from resonance Raman work?

Author Response:

Ground-state vibrational modes (i.e. stimulated Raman) have been observed in a number of clusters. In particular the work of the Goodson, Jin (and collaborators), and Moran groups on gold clusters of similar size is relevant (cited in the text). As discussed in our answer to reviewer 1, we have expanded our discussion of the nature of the vibrational mode in both Results and Discussion, and have included references to far-IR spectroscopy studies on similarly sized Zeolite-templated Ag clusters as additional support.

8) On page 15, the authors claim that the coherence at the H-L peak transferred from H to L is much stronger than the coherence directly photoexcited at L. This is different from other systems in which the direct excitation of coherence in either ground or excited electronic states would be expected to be stronger than a coherence transfer from another states. In addition, the vibrational coherence in the ground electronic state is typically stronger than that in excited electronic state. The authors should comment on this in the manuscript.

Author Response:

We have expanded and clarified our discussion of this in the text (pages 11-12).

A brief qualitative explanation is as follows: the displacement (and thus Huang-Rhys factor) of the H-band states appear to be quite small, giving a weak impulsive stimulated Raman signal (since there is no population in this state at 150+ fs, we observe no SE signal from the H-band after this).

On the other hand, the relative QB amplitude on direct excitation of the L-band is large (~25% of the total signal). Thus we get only a weak total QB signal by direct excitation into these states since 1) the H-R factor of the H-band is small, and 2) the total oscillator strength of the L-band (where the H-R factor is large) is small.

Finally: the total signal at the H-L crosspeak is large due to the large oscillator strength of the H-band where we excite, and the large H-R factor of the L-band, in sum resulting in a strong QB signal (comparable coherent/non-coherent ratio as on direct L-band excitation)

Reviewer #3 (comments to the authors):

This is a novel work that reports an exciting ultrafast multidimensional spectroscopy study of an interesting DNA/Ag cluster.

The 2D experiments presented here are very challenging. They require full phase and polarization control of four optical beams. This is definitely a state of the art experimental work and the data quality is excellent. The results are interpreted using a phenomenological three level model that allows the authors to extract energies and various relaxation rates. A major drawback is that the nature of these states is not revealed by the experiments. This will require extensive modeling and electronic structure simulations.

The paper is well written and the results have potential applications to devices and sensors. It will be of broad interest despite the limitations of the modelling. I believe that this will make a timely contribution to Nature Communications that will stimulate additional work.

Author Comments:

As the reviewer points out, the nature of quantum mechanical states is in general not directly revealed in optical experiments, but rather requires extensive and thorough theoretical work. We do expect that the experimental observables presented in this work will assist in the assembly of accurate physical models for the electronic structure and dynamics in noble metal clusters.

REVIEWERS' COMMENTS:

Reviewer #1 (Remarks to the Author):

The revised manuscript sounds better than before, and most questions are addressed reasonably, I thus recommend for publication.

Reviewer #2 made confidential comments to the editor only.